# Porous Diatomaceous Earth/Nano-Zinc Oxide Composites: Preparation and Antimicrobial Applications

Chin-Chun Chung [1] and Jiunn-Jer Hwang [1,2,*]

1   Department of Chemical Engineering, Army Academy, Chung-Li District, Taoyuan 320316, Taiwan
2   Center for General Education, Chung Yuan Christian University, Chung-Li District, Taoyuan 320314, Taiwan
*   Correspondence: jiunnjer1@gmail.com or jiunnjer@aaroc.edu.tw

**Abstract:** This paper presents the preparation and characterization of a porous and antimicrobial composite material consisting of diatomaceous earth, an inorganic pore-forming agent, and nano-zinc oxide (ZnO). A modified direct precipitation device produced high-surface area ZnO powder. The effect of reaction temperature, volume flow rate, and titration rate on ZnO particle size was studied. Using sodium chloride, potassium nitrate, and sodium percarbonate as porosity to create porous structures through dissolution was also investigated. This study found that adding cement sand to diatomaceous earth improved mold strength while lowering the volume flow rate, and increasing the reaction temperature increased the specific surface area of ZnO. At 60 °C, the crystalline structure changed from an irregular spherical form to a regular nanorod structure. The specific surface area of the prepared ZnO nanorods reached over 15 $m^2/g$, which is about five times higher. In an antibacterial experiment, adding 5% ZnO nanorods of 50 nm diameter to the porous diatomaceous earth composite material resulted in a nearly 100% antibacterial rate against *E. coli* in an aqueous environment. The results suggest that the porous diatomaceous earth/nano-ZnO composite has potential applications as an antimicrobial material, and the modified direct precipitation method could have broader implications in materials science.

**Keywords:** diatomaceous earth; nano-zinc oxide; porous nanocomposites; antimicrobial; nanorod

## 1. Introduction

Nobel's discovery in the 19th century that diatomaceous earth could safely adsorb nitroglycerine has led to its modern day use in moisture control for building materials. Diatomaceous earth is a biogenic sedimentary rock made up of the remains of diatoms with various shapes, such as disc-shaped, needle-shaped, cylindrical, and plume-shaped [1,2]. Its composition mainly consists of $SiO_2$, but contains other components such as $Al_2O_3$, $Fe_2O$, CaO, MgO, and other organic substances [3–6]. Due to its microporous structure, high porosity, and large specific surface area [7], diatomaceous earth possesses unique properties such as solid adsorption [8–10], light weight, sound insulation, solidity, wear, and acid resistance. As such, it has become an important raw material for the building, environmental, and fine chemical industries [11–17].

On the other hand, ZnO is an oxide of zinc that is insoluble in water but soluble in acids and strong alkalis. It is a white solid, also known as zinc white, obtained by burning zinc or roasting sphalerite (zinc sulfide). ZnO is used as an additive in various materials and products, such as an accelerator for rubber, pigment [18], and ceramic additives in industry [19,20], and as UV absorbers [21], nanocatalysts [22–24], lithium batteries [25,26], solar cells [27,28], and gas sensors for optoelectronic applications [29–31]. When reduced to nanosize, ZnO powder becomes a new type of multifunctional material that changes surface electronic and crystalline structures. The antibacterial mechanism of ZnO is divided into the following. (1) Photocatalytic mechanism. That is, under the irradiation of sunlight, incredibly ultraviolet light, in water and air, nano-ZnO can release

negatively charged electrons ($e^-$) by itself while leaving behind positively charged holes ($h^+$), which stimulates the generation of active oxygen by air, and the active oxygen can oxidize with a variety of microorganisms, thus achieving the bactericidal effect [32,33]. (2) Metal ion dissolution mechanism. In other words, the free zinc ions bind to proteases when they come into contact with bacteria and inactivate them to achieve the bactericidal effect [34–36]. In a quantitative bactericidal experiment on nano-ZnO, it was found that when the concentration of nano-ZnO was 1%, the bactericidal rate of Staphylococcus aureus was 98.186% and the bactericidal rate of *E. coli* was 99.19% within 5 min [37–42].

A porogenic agent is a substance added to a powder mixture whose purpose is to form the required type and number of pores in the final product through its volatilization during sintering, and it must meet the characteristics of easy elimination during heating, no harmful residues in the matrix after elimination, and does not react with the matrix [43–45]. Pore-forming agents in making porous materials can be classified as organic and inorganic. Organic pore-forming agents include natural fibers, polymers, and organic acids, while inorganic pore-forming agents include ammonium carbonate, ammonium bicarbonate, and ammonium chloride. Organic pore-forming agents oxidize during sintering at high temperatures and leave pores. In contrast, complete thermal decomposition at high temperatures can eliminate inorganic porous agents, leaving no residual material in the pores. Ideal porogenic agents should also have controllable pore size and uniform distribution, mainly used in industry to make nano-grade and tiny irregular pores [46–48].

In this experiment, we aim to investigate the factors that affect the surface area of nano-ZnO through the co-precipitation method and the antibacterial rate of different particle sizes of ZnO against *E. coli*. We will also add nano-ZnO and inorganic porogenic agents to commercial diatomaceous earth with a particle size of about 325 mesh (44 μm) to create a high-strength, high-porosity antibacterial composite material with low energy consumption and simple operation. Furthermore, the resulting material will improve the application value of the material.

## 2. Experimental Section

### 2.1. Chemicals and Materials

We used diatomaceous earth (DE), particle size ~44 μm (Alfa Aesar, Lancashire, UK), calcium sulfate hemihydrate ($CaSO_4 \cdot 0.5H_2O$) (J.T. Baker, Randor, PA, USA), tricalcium silicate ($3CaO \cdot SiO_2$) (Sigma-Aldrich, St. Louis, MO, USA), sodium percarbonate ($2Na_2CO_3 \cdot 3H_2O_2$) (Fisher Scientific, Fair Lawn, NJ, USA), ethanol ($C_2H_5OH$) (Mallinckrodt Chemicals, Hazelwood, MO, USA), zinc chloride ($ZnCl_2$) (Thermo Fisher Scientific, Waltham, MA, USA), potassium hydroxide (KOH) (Honeywell Riedel-deHaen, Seelze, Germany), nutrient broth (Cat. No.: 213000, Becton, Dickinson and Company, Sparks, MD, USA), and polydimethyl siloxane (PDMS, Sylgard 184 A and B, Dow Corning, Midland, MI, USA). Organism *E. coli* (BCRC 10239) was obtained from the food industry research and development institute of the bioresource collection and research center (Hsinchu, Taiwan).

### 2.2. Instrumentation

We used a vacuum oven (Model: VWR 1415M, Sheldon Manufacturing Inc., Cornelius, OR, USA), controlled-temperature magnetic stirring hotplate (Model: IKA RET Basic, IKA Works, Wilmington, NC, USA), spectrophotometer (Model: Jasco V-530, Jasco Inc., Easton, PA, USA), high-temperature furnace (Model: Vulcan 3-400HTA, NEYTECH, Bloomfield, CT, USA), metallurgical microscope (Model: BX51, Olympus Corporation, Tokyo, Japan), surface area analyzer (Gemini VII 2390, Micromeritics Instrument Corporation, Norcross, GA, USA), scanning electron microscope (SEM) (Model: JSM-7600F, JEOL Ltd., Tokyo, Japan), and X-ray diffractometer (Model: D8 Advance, Bruker, Karlsruhe, Germany). All reagents were of reagent grade unless otherwise stated. *E. coli* that fluoresces when stimulated by UV light was observed using a fluorescent microscope (Model: Nikon TE2000-U, Japan) and the filter used for the autofluorescence detection of *E. coli* was UV (330~380 nm, Nikon).

*2.3. Experimental*

The experimental design is divided into three parts. In the first part, inorganic substances are added to increase the modulus of the mixture and produce a porous diatomaceous earth composite material. A high-surface area ZnO antibacterial material is prepared in the second part. In the third part, the antibacterial properties of the porous diatomaceous earth composite material are tested. The experimental setup is illustrated in Scheme 1.

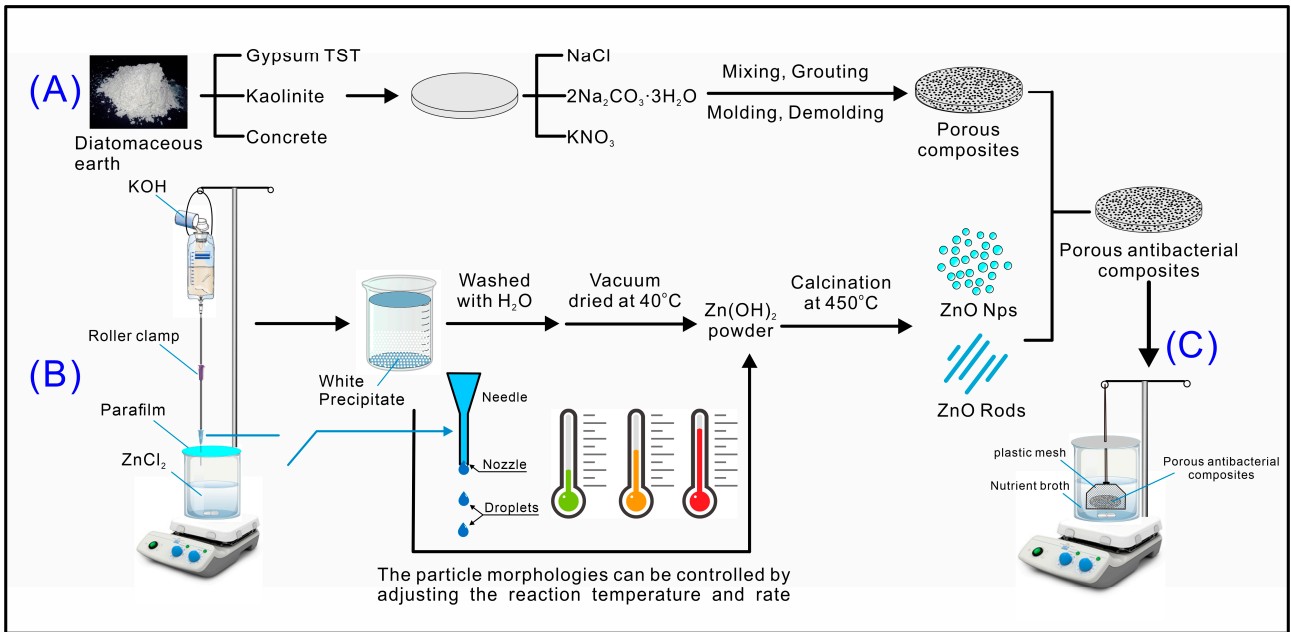

**Scheme 1.** Material preparation process: (**A**) adding inorganic materials to increase the strength of the mold and produce porous DE composite materials; (**B**) preparing high-specific surface area ZnO antibacterial materials. (**C**) The antibacterial experiment of porous DE/ZnO composites in the aqueous phase.

2.3.1. Diatomaceous Earth Mold

We prepared cylindrical plastic cups as mold tools for diatomaceous earth (DE) molding and coated the cup walls with Vaseline for easy demolding. As shown in Scheme 1A, we precisely weighed DE, TST gypsum (CaSO$_4$·2H$_2$O), and white cement powder according to the ratio, mixed the powders evenly, and added 1x volume of water to the mixture. We poured the mixture into the mold and gently tapped the bottom of the container to release gas bubbles and level the liquid surface. We left the mold to dry naturally at room temperature. After the DE composites were completely dry, we gently tapped the cup walls to de-mold and removed the DE composites. Furthermore, a pore-forming experiment was conducted using the dissolution method. The pore-forming additives included refined salt (NaCl) particles, potassium nitrate (KNO$_3$) particles, and sodium percarbonate (2Na$_2$CO$_3$·3H$_2$O$_2$) particles that produce O$_2$ upon contact with water. The DE, cement, and TST gypsum weight ratios were 4:6:3 to obtain good mold strength. The three pore-forming agents were added and compared.

2.3.2. Direct Precipitation Synthesis of ZnO Powder

We weighed zinc chloride (ZnCl$_2$) and potassium hydroxide (KOH) accurately and prepared 0.1 M and 0.2 M 1000 mL aqueous solutions, respectively. As shown in Scheme 1B, we poured the prepared KOH solution into a feeding bag and placed the ZnCl$_2$ solution in a 2000 mL beaker. We slowly dripped the KOH solution into the beaker at drop rates of 0.0125, 0.020, 0.035, and 0.065 mL/s while stirring continuously at a constant temperature (25, 40, 60, 80 °C) and 800 rpm stirring speed. After the titration was completed, we removed the upper layer of liquid, leaving the precipitate, and washed it repeatedly with

distilled water to remove $K^+$ and $Cl^-$. The washing solution can be tested for residual $Cl^-$ using a silver nitrate. We placed the precipitate in a vacuum oven (vacuum of about 500–600 mmHg) and dried it at 40 °C to obtain white zinc hydroxide ($Zn(OH)_2$) powder. The $Zn(OH)_2$ powder can be converted to white ZnO powder by calcining it at 450 °C for 4 h.

The above description is the formula of the composite material and the production process of this experiment. In order to achieve uniform distribution of inorganic TST gypsum, $2Na_2CO_3 \cdot 3H_2O_2$, and ZnO in DE-based composites, the four powders were packed into a tightly sealable plastic can and operated continuously for 30 min using V-shaped blending equipment to ensure adequate blending of the powders for subsequent experiments.

*2.4. Surface Area Measurement and BET Particle Size Calculation of ZnO Powder*

The ZnO sample was degassed at a high temperature under an inert gas ($N_2$) atmosphere. The analysis was then performed using a surface area analyzer to obtain data such as the Brunauer–Emmett–Teller specific surface area ($A_{BET}$), pore volume, and pore size of the ZnO powder. The BET particle size ($d_{BET}$) can be calculated using the following equation with the measured surface area.

The $d_{BET}$ is an average surface diameter used to describe the size of particles, assuming they are spherical and monodisperse. It is calculated from the specific surface area measured using the Brunauer–Emmett–Teller (BET) method [49]. The formula for calculating $d_{BET}$ is as follows:

$$d_{BET} \text{ [nm]} = 6000/(A_{BET} \times \rho)$$

where $A_{BET}$ = BET specific surface area $[m^2/g]$ and $\rho$ = theoretical density $[g/cm^3]$.

*2.5. Antibacterial Effect of the Porous DE/Nano-ZnO Composites*

We prepare four 250 mL beakers and placed a stir bar at the bottom. We added 150 mL of culture medium solution to each beaker. We loaded the pre-formed porous DE composite material into plastic mesh filter bags and suspended them at the top of each beaker to avoid direct contact with the medium culture solution. Then, we sealed the beaker openings with aluminum foil and placed them on a hotplate. For sterilization, we boiled the culture medium and diatomaceous earth composite material continuously at 100 °C for 30 min. Generally, the higher the temperature, the shorter the required sterilization time. Sterilization can be achieved by processing materials in saturated high-pressure steam at 121 °C for 15 min. However, considering the purpose of this study, the sterilization of nutrient broth (NB) medium and composite materials was performed using a magnetic stirrer to directly heat and boil them continuously for 30 min. The NB medium solution was subjected to these sterilization conditions and left at room temperature for 3 days. The changes in absorbance of the culture medium during this period were examined using a spectrophotometer, and the experimental results confirmed that this heating method can also achieve good sterilization efficacy. Once the medium culture solution had cooled to room temperature, we immersed the suspended diatomaceous earth composite material into the medium culture solution, as shown in Scheme 1C.

We used the culture's optical density ($OD_{600nm}$) as a quantitative measure of bacterial growth to evaluate the antibacterial activity of the porous DE composite material against *E. coli*. We set up a control group (DE composites) and an experimental group (DE/nano-ZnO composites) to compare bacterial growth rates and behavior. First, we inoculated the growth medium with overnight culture and ensured the initial cell OD values were within a reasonable range (0.045–0.06). Then, we injected 0.5 mL of a fixed-concentration *E. coli* solution into each beaker using a syringe. We mixed the *E. coli* solution with the DE composite material during the antibacterial test and stirred at 300 rpm to ensure complete contact and mixing. We repeated the experiment three times to ensure the reliability of the results. We measured the $OD_{600nm}$ values of the cultures every hour for 10–12 h and took a final reading at 24 h [50].

The antibacterial rate (%) is calculated as $[(A - B)/A] \times 100\%$, where A is the absorbance ($OD_{600nm}$) of the control group and B is the absorbance ($OD_{600nm}$) of the experimental group. The antibacterial rate results help assess the differences in antibacterial effectiveness when the DE composite material is enhanced by adding nano-ZnO.

## 3. Results and Discussion

### 3.1. Comparison of Adding Inorganic Materials to Diatomaceous Earth for Mold Making

Diatomaceous earth (DE) powder has extremely high plasticity but no inherent viscosity. Therefore, the product made from DE after drying lacks strength and is easily destroyed by external forces, thus limiting its material processing characteristics and application value. In a gas-phase environment, the strength and durability of the product can be improved by adding organic or inorganic materials. However, when applying it to a liquid-phase climate, in addition to considering the product's durability in water, the porosity of the DE after molding must also be increased to allow the internal pores of the material to be fully utilized. Figure 1a shows that when DE is mixed with an appropriate amount of water, it is easy to crack and cannot be molded alone after drying. To enhance the processing performance of DE, this experiment aimed to investigate the effect of incorporating TST gypsum, kaolin, and white cement powder in a 1:1 weight ratio. These materials were mixed with suitable water (powder:water = 1:1), and the resulting mixture was used for drying and molding comparison. Figure 1b,d shows that DE can be successfully demolded after adding TST gypsum powder and white cement powder, and there is almost no powderization problem. Therefore, this study used TST gypsum and white cement powder as the primary materials for improving the strength of DE.

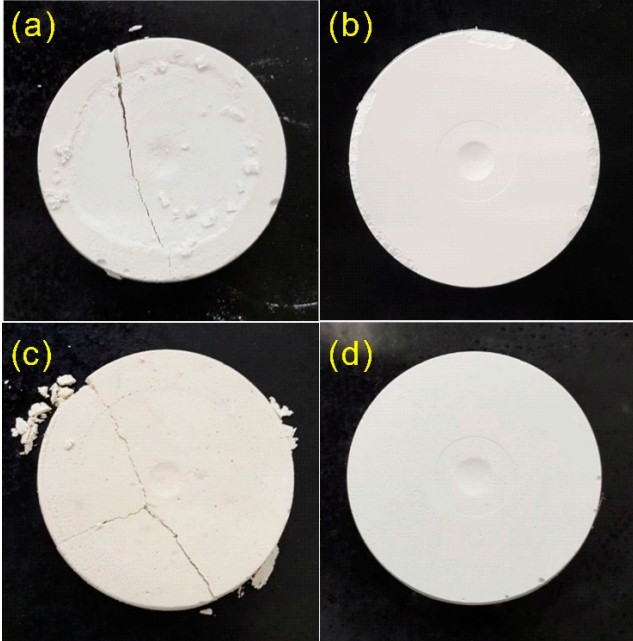

**Figure 1.** Shows the formula for the DE composites (DE:inorganic material) = (5:5). The appearance of the DE composites after adding inorganic materials for molding is shown as follows: (**a**) without adding anything, (**b**) TST gypsum, (**c**) kaolin, and (**d**) white cement powder. Almost no powdering problem exists in (**b,d**).

The cement composition contains substances such as tricalcium silicate ($C_3S$) and dicalcium silicate ($C_2S$) that have cohesive properties. When mixed with water, it immediately undergoes hydration, producing various hydration products and crystals. Therefore, it is quite suitable as an inorganic additive for increasing the strength of DE. Five ratios of DE:white cement, 7:3, 6:4, 5:5, 4:6, and 3:7, were compared for solidification, and the experimental results are shown in Figure 2. All five DE formulations can be successfully

dried and molded. However, although the DE in Figure 2b,f can be molded, the powder can still be scraped off from the surface after demolding, indicating that the product's strength is insufficient. This can cause the DE to disintegrate in a liquid-phase environment and cause water pollution quickly. On the other hand, the DE molded in Figure 2c–e can be successfully demolded and has no powderization problem in appearance. To maintain sufficient strength and water absorption properties of the composite material, a DE:white cement powder ratio of 4:6 was selected as the optimal ratio for the subsequent experiments in making DE composite materials.

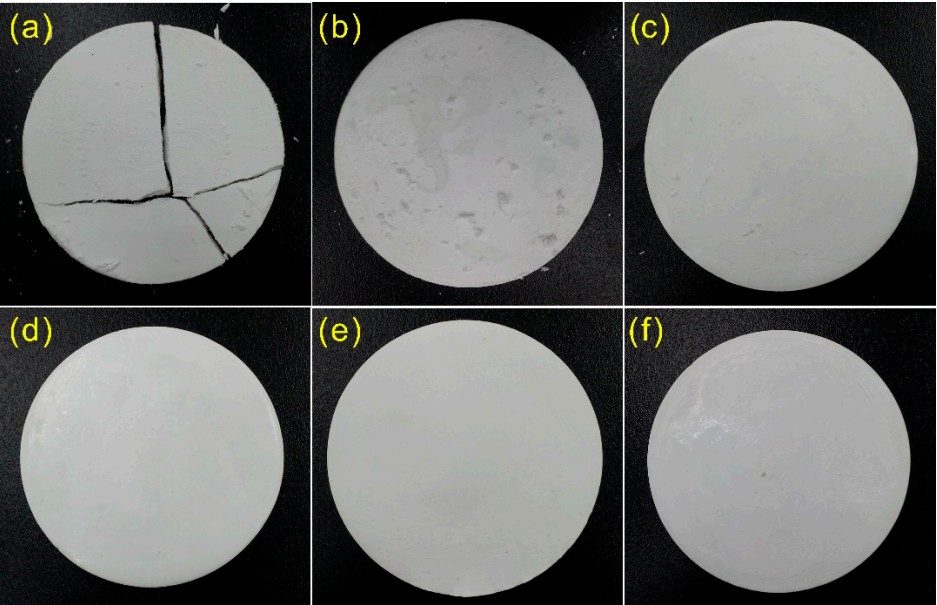

**Figure 2.** The schematic diagrams of different ratios of DE and white cement for making molds: (**a**) without adding DE; (**b**) 7:3; (**c**) 6:4; (**d**) 5:5; (**e**) 4:6; (**f**) 3:7. The DE after molding in (**c**–**e**) can not only be successfully de-molded but also has no powdering issue on the surface.

Due to the extended surface drying time of DE mixed with cement, which typically exceeds 5 h, and the fact that the surface cannot be disturbed during the drying period to avoid surface unevenness and even the destruction of the mold structure's strength, it is necessary to improve the drying rate of DE material. TST gypsum, hydrated calcium sulfate ($CaSO_4 \cdot 2H_2O$) with water, can be used as a cement retarder to control its setting rate. Therefore, an attempt was made to enhance the drying rate of DE by adding TST gypsum. DE and cement were mixed in a ratio of 4:6, and 0, 10, 20, 30, and 40% TST gypsum were added separately. As shown in Figure 3, the experimental results were obtained after two repeated experiments. It was found that DE took approximately 318 min to dry on the surface without adding TST gypsum. However, when 10% TST gypsum was added, the drying time was shortened to 87 min, approximately 0.27 times the original time. When the amount of TST gypsum was increased from 20% to 40%, entire surface drying could be reached within approximately 1 h. From the trend of the graph, it was observed that the proportion of TST gypsum added was linearly inversely proportional to the drying time. When DE was mixed with cement and encountered water, it reacted with the water. Most of the water in the DE and cement mixture did not evaporate but combined with cement to form a hydrate. The strength of the hydrate increased with time. Therefore, too much TST gypsum should not be added to avoid too fast a drying rate, which would affect the cement's hydration reaction and reduce the strength of DE after molding. To shorten the drying time to within 1 h, the appropriate amount of TST gypsum to be added ranged from 20–30%. The related experiments were performed using an evenly mixed powder (DE composite material) of DE/cement/TST gypsum.

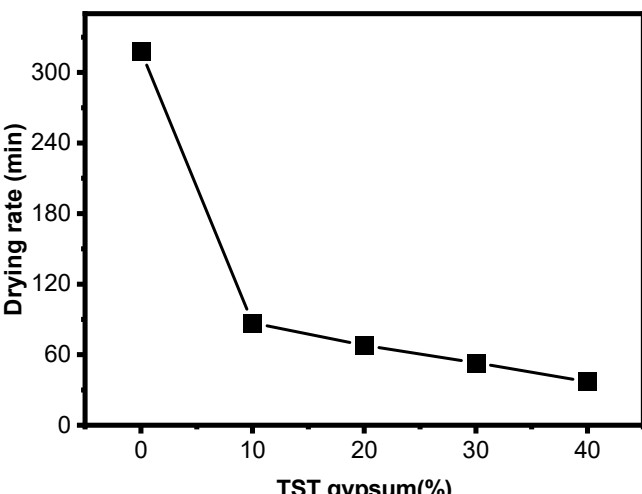

**Figure 3.** Illustration of the effect of adding different ratios of TST gypsum to DE/cement powder (4:6) composites on the drying rate.

### 3.2. Comparison of Porosity of Diatomaceous Earth Composites

In the existing technology, porous ceramic products or inorganic composite materials usually add pore-forming agents to the formulation. Porous products can be obtained through mixing, molding, drying, and firing. The pores are formed because these pore-forming agents burn or evaporate at high temperatures. However, the high-temperature combustion process usually produces harmful gases, causing pollution to the atmosphere. In this study, after multiple attempts at different pore-forming methods mentioned in the literature, the dissolution method was chosen for the experiment. The pore-forming agents were easily obtainable and highly soluble fine salt particles, potassium nitrate ($KNO_3$) particles whose solubility changes drastically with temperature, and sodium percarbonate ($2Na_2CO_3 \cdot 3H_2O_2$) particles that produce $O_2$ when in contact with water. The weight ratios of the previous DE:cement:TST gypsum with good molding strength were 4:6:3. Three pore-forming agents were added and compared. The experimental results are shown in Figure 4a–d. The DE without any pore-forming agent added has a smooth and intact structure on the surface, and almost no pores can be seen with the naked eye. However, when a small amount of acceptable salt is added to the DE after drying and washing, tiny pores left by the fine salt particles can be observed on the surface. When the amount of fine salt particles added increases to 20 g, an evenly distributed pore structure can be obtained on the surface of the DE, as shown in Figure 4b. When the DE is cut open after molding, the cross-section can also reveal the pore structure formed through the dissolution of fine salt particles. These tiny pores are slightly smaller than the diameter of the fine salt particles. It is presumed that most of the fine salt particles added to the DE composite material partially dissolved during the mixing process with water.

The second pore-forming agent tested was $KNO_3$. Due to its low solubility in water at low temperatures (13.3 g$KNO_3$/100 g $H_2O$ at 0 °C), the DE composite material was prepared by adding $KNO_3$ and then setting the mold in ice-cold water at 4 °C. This reduced the solubility of the foaming agent in water. After drying the mold, warm water was used to dissolve the $KNO_3$ inside the DE composites to create pores. The appearance of the molded sample after this process is shown in Figure 4c. Although pores were formed on the surface of the DE composites after washing, the pore size distribution was quite uneven. When the amount of $KNO_3$ added increased to 3 g, there was partial detachment on the surface, and the original structure of the DE composites became fragile and loose due to the addition of $KNO_3$. Since the surface pores were sparse and uneven, it was not suitable for use as a foaming agent.

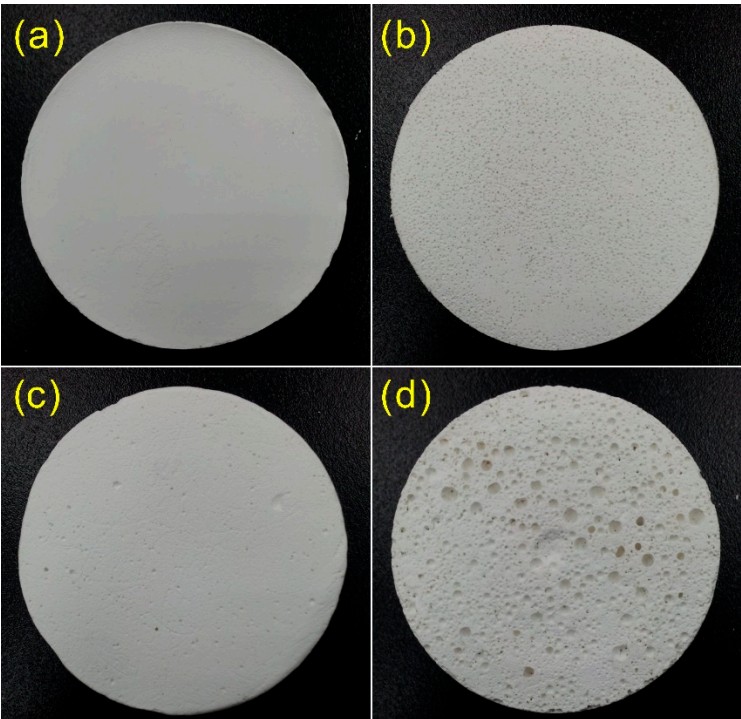

**Figure 4.** DE composites with different porogenic agents after washing: (**a**) no addition; (**b**) 20 g NaCl; (**c**) 3 g $KNO_3$; (**d**) 3 g $2Na_2CO_3\cdot3H_2O_2$ (Note: DE composites' formulation is DE:cement:TST = 4:6:3).

The third pore-forming agent used was $2Na_2CO_3\cdot3H_2O_2$ particles. These particles rapidly produce unstable hydrogen peroxide upon contact with water, which quickly decomposes to form oxygen and water. Therefore, the amount of water used to mix with the DE composites after adding the foaming agent must be carefully controlled. Excess water will cause the foaming agent dispersed in the DE composite material to dissolve prematurely and produce a large amount of $O_2$, causing excessive material expansion. Although the dried DE composites had many pore structures, the structural strength decreased significantly. Therefore, appropriate material and water proportions are critical to achieving foaming and stability. The experimental results are shown in Figure 4d. When the amount of water used was just enough to mix the powder, not only did the $2Na_2CO_3\cdot3H_2O_2$ particles dispersed in the DE composites dissolve to produce a large number of pores, but a portion of the particles also remained undissolved. When these particles were dissolved in water, pore structures were formed, and the internal structure of the finished product resembled a porous cheese structure with a broader distribution of pore sizes.

After perforation, the composite material of DE appeared to have small holes on the surface visible to the naked eye. However, further confirmation of the microscopic structure of the material's surface using a metallographic microscope helped understand the effects of using different perforation agents on the DE composite material. Figure 5a–d depicts images of the DE composites with varying agents of perforation added, as captured under a metallographic microscope. Figure 5a shows that when observed under a microscope at $100\times$ magnification, the surface of the DE composite material remains flat and almost without any pores. However, after the addition of perforation agents, except for the $KNO_3$ group, which has sparse pores on the overall surface, the finished product surface of DE with the addition of the perforation agents sodium chloride (NaCl) and $2Na_2CO_3\cdot3H_2O_2$ has distributed pores, as shown in Figure 5b–d. When using NaCl as a perforation agent for the DE composite material, many pores were left after the remaining NaCl particles were removed by washing or soaking the material in water. Although the shape of the pores is irregular, their size distribution is relatively uniform.

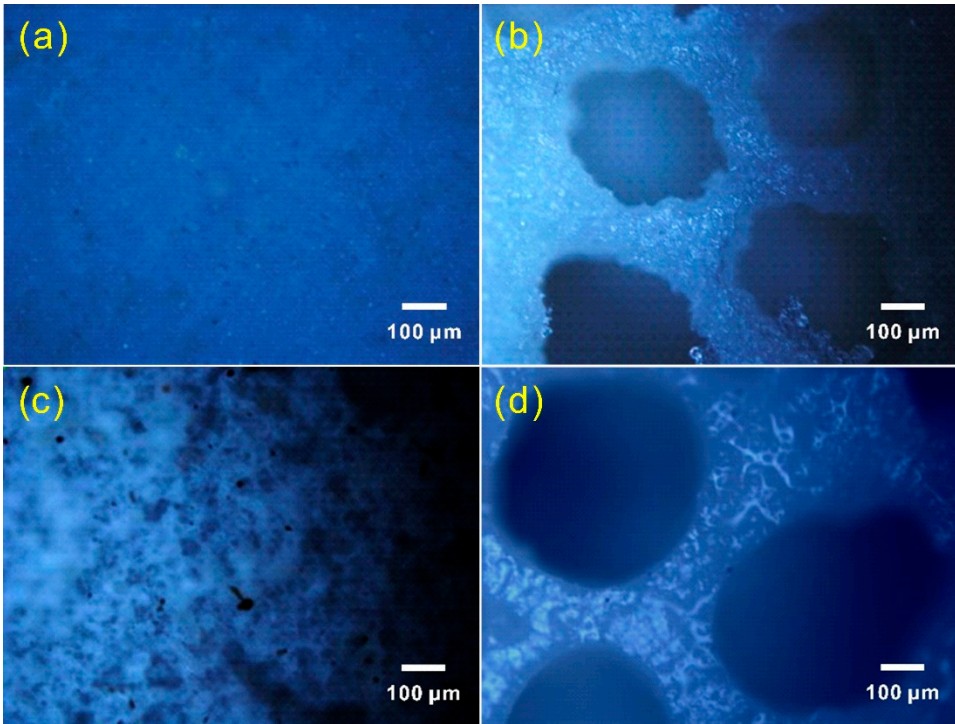

**Figure 5.** Metallographic images (100×) of DE composites with different porogenic agents after washing: (**a**) no addition; (**b**) 20 g NaCl; (**c**) 3 g $KNO_3$; (**d**) 3 g $2Na_2CO_3 \cdot 3H_2O_2$ (Note: DE composites' formulation is DE:cement:TST = 4:6:3).

In contrast, when $KNO_3$ was used as the perforation agent, not only was the distribution of the pores highly uneven, but the overall size was also much smaller than those formed by using NaCl. When $2Na_2CO_3 \cdot 3H_2O_2$ was used as a agent for the DE composites, the surface presented many pore structures that were nearly spherical. The size of these pores was irregular, with large pores primarily resulting from bubbles formed by the reaction between $2Na_2CO_3 \cdot 3H_2O_2$ and water and left behind after drying. At the same time, the rest came from the pores left by the unreacted $2Na_2CO_3 \cdot 3H_2O_2$ particles after dissolution in water through washing. As a result, the pores on the material surface were unevenly distributed. Adding NaCl to the DE composite material during the perforation process produced smaller pores with a uniform distribution. To increase the pore size, NaCl may be replaced with coarser salt. In addition, the solubility of potassium nitrate ($KNO_3$) in water is greatly affected by temperature. When perforating at 4°C with cold water, some $KNO_3$ particles still dissociate into $K^+$ and $NO_3^-$, while most of the $KNO_3$ particles can remain inside the material during the perforation process, making the surface of the DE brittle after molding, making it prone to peeling during washing, which significantly limits the possibility of using DE in water. Therefore, to increase the porosity inside the DE composite material, $2Na_2CO_3 \cdot 3H_2O_2$ is a preferable perforation agent.

Both sodium chloride and sodium percarbonate can be used as pore-forming agents for diatomaceous earth composites. Removing the pore-forming agent dispersed in the composite material using a dissolution method is a simple and environmentally friendly method. If sodium chloride is used as the pore-forming agent, the pore structure in the composite will be related to the size of the mixed sodium chloride crystals. Strictly controlling the solubility of sodium chloride in water can retain the residual sodium chloride crystals in the composite material. Immersing the diatomaceous earth composites in hot water can dissolve the sodium chloride crystals and naturally form a large number of uniform porous structures without affecting the original strength of the composite material. On the other hand, when sodium percarbonate is used, it releases a large amount of oxygen when it comes into contact with water. Since the oxygen is encapsulated in the

diatomaceous earth composites, the originally dense structure will form a loose structure due to the encapsulation of a large amount of oxygen. The formation of pores comes from the oxygen encapsulated in the composite material and the remaining sodium percarbonate particles, and the pore size is less uniform.

*3.3. Preparation of High-Surface Area ZnO Powder*

This experiment used a simple chemical synthesis method to prepare high-surface area ZnO powder. This method dramatically reduces the cost of raw materials and produces ZnO powder with a high surface area by changing the experimental parameters, enhancing its antibacterial application. The experiment was based on the direct precipitation method, which improved the equipment threshold required for traditional chemical reactions, making it possible to produce high-yield and high-surface area ZnO powder with simple and inexpensive reaction equipment. According to the literature [51–53], if 0.1 M $ZnCl_2$ and 0.2 M KOH are used for direct precipitation, high-surface area ZnO powder can be produced. Therefore, this study explored the effect of changing the titration rate, reaction temperature, and volume flow rate on the product surface area by using this concentration to find the experimental process for producing high-surface area nano-ZnO.

First, a 0.2 M KOH solution was filled into a feeding bag, and a roller clamp controlled the liquid flow rate. Next, the solution was slowly dripped into the vigorously stirred $ZnCl_2$ water solution at room temperature. The final deposit obtained through the direct precipitation method was a white powder suspended in water. When the particle size of the suspended particles is small enough, the energy of their Brownian motion is sufficient to prevent the effect of gravity, thereby preventing sedimentation and maintaining a stable state for a long time. After removing the upper layer of water, the remaining white deposit in the beaker was dried under a vacuum to remove residual moisture, and fluffy white powder was obtained. The products were dried at 40 °C and under vacuum conditions to avoid secondary aggregation of the deposits at low temperatures. All these low-temperature dried products were chemically composed of $Zn(OH)_2$.

White $Zn(OH)_2$ is an amphoteric hydroxide compound that decomposes into ZnO and water above 125 °C. The white deposits prepared through the direct precipitation method must be converted entirely into ZnO powder using high-temperature calcination to prepare nano-sized ZnO powder. The calcination temperature of the intermediate product $Zn(OH)_2$ is a critical factor. If the intermediate product is treated at 400 °C, it requires a longer heating time, but the particle size of the ZnO will be severely affected under excessively high-temperature conditions. Therefore, this experiment used the conditions of 450 °C and 4 h to treat the $Zn(OH)_2$ intermediate product. The yield obtained through the precipitation method can be calculated by weighing the sample after cooling, as shown in Table 1. At room temperature, the ZnO yield was determined using a small needle with drop rates of 0.5, 1, 2, and 3 drops/s, resulting in yields of 89.03%, 89.47%, 89.85%, and 87.45%, respectively. All four drop rates produced yields of 87.45% or higher, with slight differences attributed to losses during the washing process and incomplete reaction. The specific surface area of the powder measured using a BET analyzer ranged from 1.9543 to 2.4754 $m^2$/g. It was initially believed that the powder obtained at a drop rate of 0.5 drops/s would produce a higher yield and specific surface area of $Zn(OH)_2$ powder with a more porous structure. However, the experimental results obtained through specific surface area analysis did not meet expectations. It is postulated that this may be due to the large amount of water vapor that boils and rapidly evaporates onto the surface of the deposited material in the vacuum drying of the $Zn(OH)_2$ aqueous solution. The unevaporated water vapor inside the deposited material causes it to expand instantly, resulting in a sponge-like structure under vacuum pressure. Further application of external force will break up the porous structure and generate fine $Zn(OH)_2$ powder, which can be converted to fine and uniform white ZnO powder after calcination. The experimental results suggest that reducing the drop rate of the reactant at room temperature seems more conducive to increasing the specific surface area of ZnO.

**Table 1.** Effect of different drop rates of the small needle on the yield and specific surface area of ZnO at room temperature.

| Drop rate (drop/s) | 0.5 | 1 | 2 | 3 |
|---|---|---|---|---|
| Weight (g) | 7.2449 | 7.2809 | 7.3123 | 7.1168 |
| Yield (%) | 89.03 | 89.47 | 89.85 | 87.45 |
| BET surface area ($m^2$/g) | 2.2691 | 2.4754 | 1.9543 | 2.1196 |
| $d_{BET}$ (nm) | 472 | 432 | 548 | 505 |

Note: ZnO theoretical yield weight = 8.138 g; small needle drop rate = 0.0125 mL/s.

The direct precipitation method was used to produce small-sized particles, where stirring rate and reaction temperature are essential factors controlling nucleation and crystal growth. Generally, the resulting particle size will increase if the nucleation rate is slow and the nucleus growth rate is fast. Therefore, to reduce the average particle size of the deposited material, the nucleation rate must be accelerated while the crystal nucleus growth rate is reduced. The traditional direct precipitation method uses a KOH aqueous solution mixed with a $ZnCl_2$ aqueous solution, followed by rapid stirring. Although this method can significantly reduce the reaction time, it is not easy to control the nucleation and growth of crystals when the reactants collide. In addition, although the nucleation rate can be accelerated under high-speed stirring conditions, it is difficult to suppress the growth rate of crystal nuclei.

This study used an improved direct precipitation method to produce $Zn(OH)_2$ powder. The main difference between this method and that of previous researchers is that the reaction is carried out by slowly adding the reactants dropwise. Under high-speed stirring conditions, this can avoid the problem of excessive crystal nucleus growth rate, as shown in Figure 6. Observing the volume of the product powder with the naked eye under four temperature conditions, it can be seen that although the powder yield synthesized under different temperature conditions does not vary significantly, the powder volume is quite different. In particular, at 60 °C, the amount of powder synthesized is almost four times more than that of 25 °C, indicating that the powder synthesized at high temperatures has a smaller particle size. This remarkable phenomenon caught our interest, and ZnO synthesized at different temperatures was tested using SEM. The test results are shown in Figure 7. When the temperature was maintained at room temperature (25 °C), the morphology was spherical with a particle diameter of about 200–300 nm. When the temperature increased by 40 °C, the particle diameter became smaller, about 200–100 nm. When the temperature rose above 60 °C, the synthesized ZnO changed from a spherical to a nanorod shape.

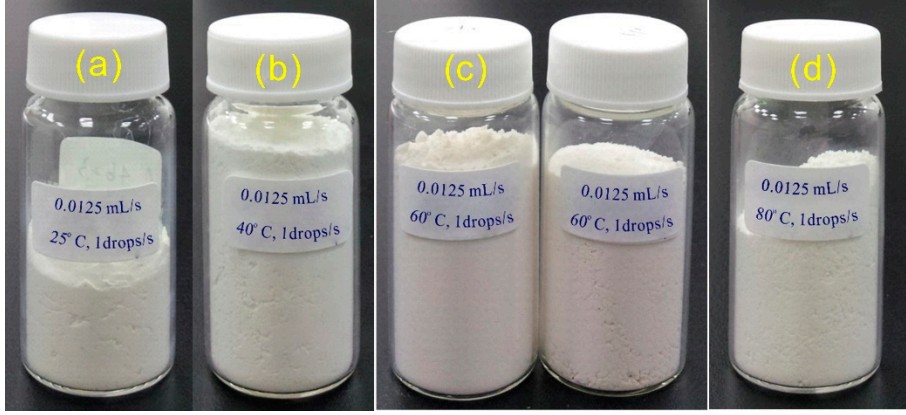

**Figure 6.** Nano-ZnO powder prepared at different reaction temperatures: (**a**) 25 °C; (**b**) 40 °C; (**c**) 60 °C; (**d**) 80 °C (the volume of ZnO powder synthesized at 60°C is almost four times more than that at 25 °C).

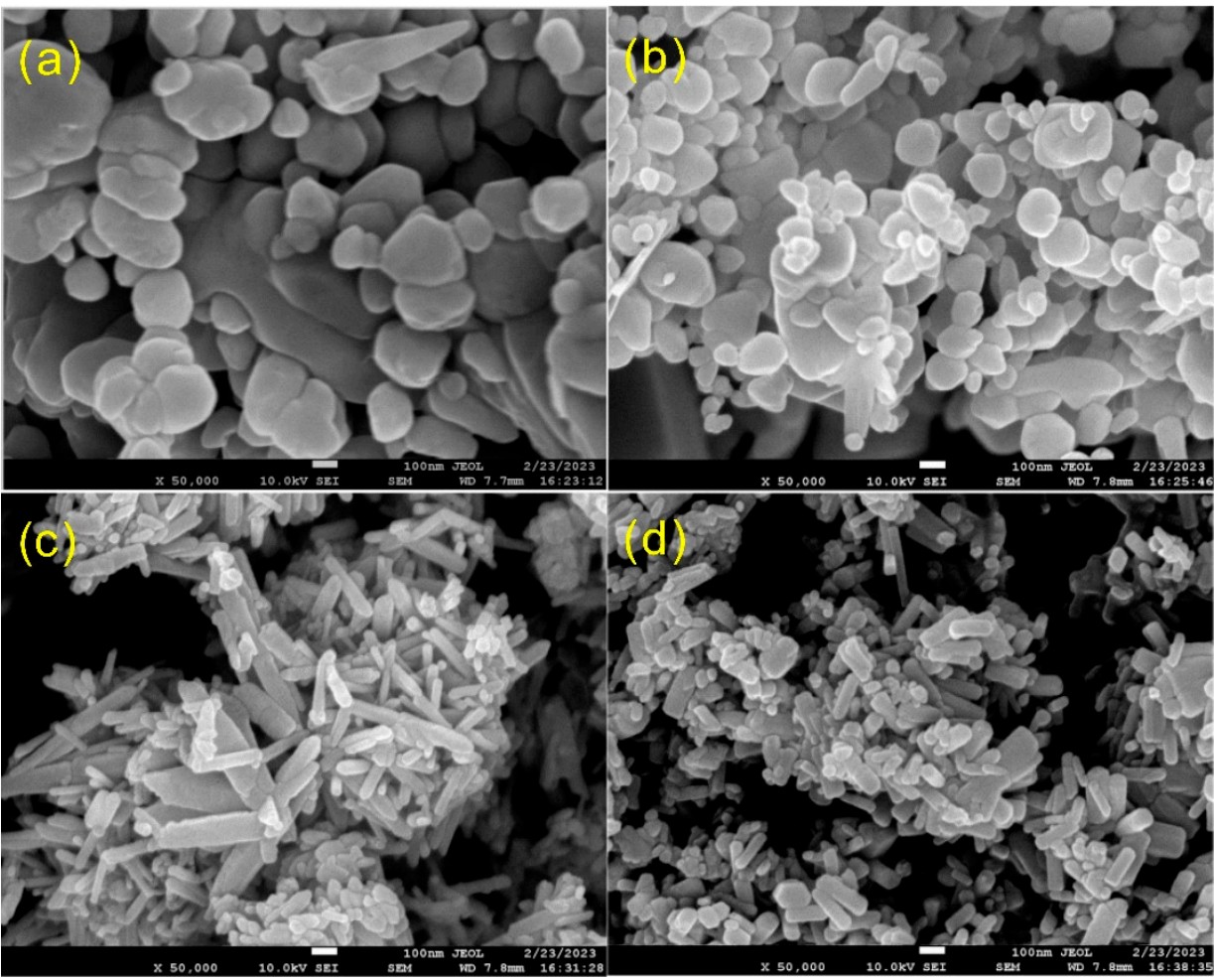

**Figure 7.** SEM images (50,000× magnification) of nano-ZnO at different reaction temperatures: (**a**) 25 °C; (**b**) 40 °C; (**c**) 60 °C; (**d**) 80 °C. In addition, the observation of crystal morphology is compared.

The BET test results, as shown in Table 2, showed that the specific surface area of the ZnO particles increased significantly with the increase in reaction temperature after calcination treatment. The specific surface areas obtained at reaction temperatures of 25, 40, 60, and 80 °C were 2.9568, 10.3577, 14.8765, and 11.3191 m$^2$/g, respectively, with corresponding BET diameters of 362, 103, 72, and 143 nm. When the reaction temperature increased from 25 °C to 60 °C, the specific surface area of the powder increased by about five times. The experimental results showed that with the increase in temperature, the reaction rate increased rapidly, and the supersaturation increased, which was conducive to the formation of a high number of nuclei, and at the same time, the growth rate of the grains slowed down, which was conducive to the steady growth of ZnO crystal morphology. The rod-shaped nanorods help improve the antibacterial effect.

**Table 2.** Comparison of the values of ZnO yield, pore volume, and specific surface area at different reaction temperatures.

| Temperature (°C) | 25 | 40 | 60 | 80 |
|---|---|---|---|---|
| Yield (%) | 89.85 | 94.84 | 89.38 | 92.01 |
| BET surface area (m$^2$/g) | 2.9568 | 10.3577 | 14.8765 | 11.3191 |
| Pore volume (cm$^3$/g) | 0.001457 | 0.005221 | 0.007403 | 0.005644 |
| $d_{BET}$ (nm) | 362 | 103 | 72 | 143 |

The ZnO nanorods prepared at four different temperatures were subjected to XRD testing, and the results, as shown in Figure 8, indicate the appearance of characteristic absorption peaks of the ZnO crystal structure, with 2θ and lattice planes of 31.76° (100), 34.42° (002), 36.28° (101), 47.54° (102), 56.67° (110), 62.94° (103), 66.39° (200), 67.92° (112), 69.08° (201), 72.52° (004), and 76.96° (202). These results confirm that this study's experimental method can synthesize ZnO material with a nanostructure using a low-cost and straightforward direct precipitation method. However, when the reaction temperature reached 60 °C and 80 °C, the structure showed a more significant amount of nanorod structures. In addition, the full width at half maximum (FWHM) of the characteristic absorption peak of ZnO became slightly broader.

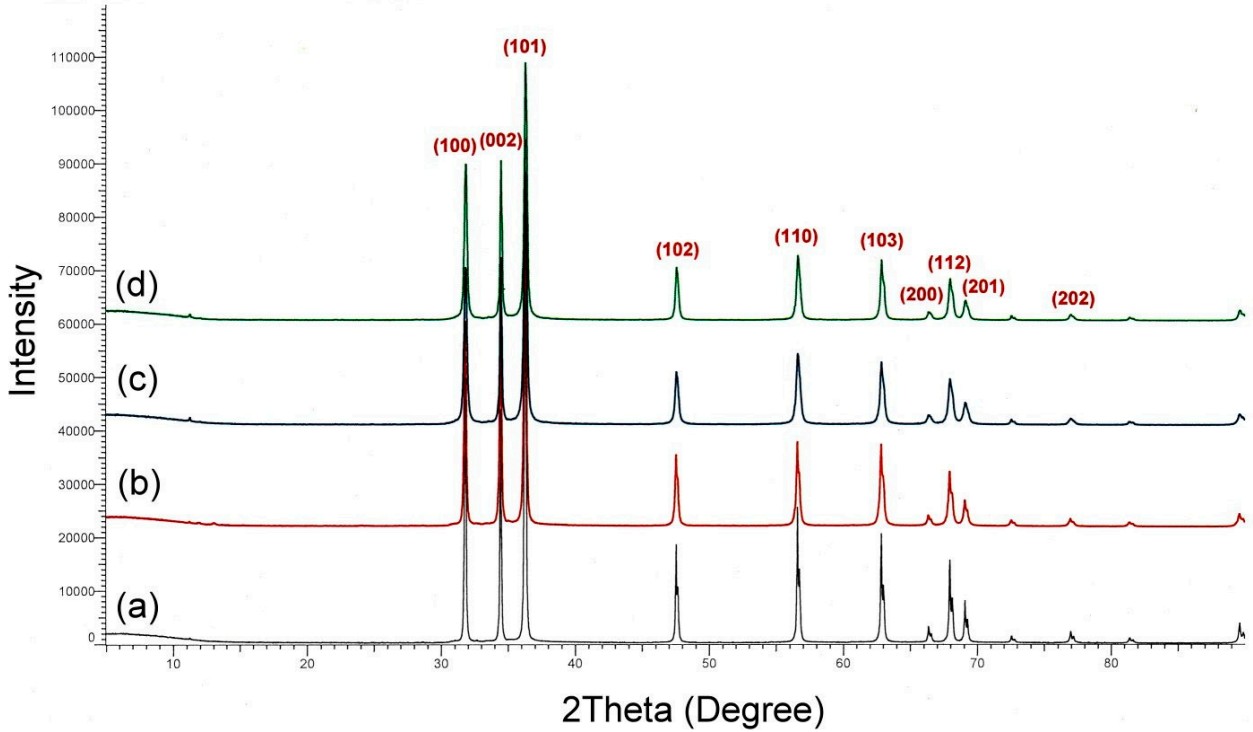

**Figure 8.** XRD crystal diffraction analysis of nano-ZnO at different reaction temperatures: (**a**) 25 °C; (**b**) 40 °C; (**c**) 60 °C; (**d**) 80 °C.

According to the previous experimental results, although using direct precipitation to enhance the reaction temperature significantly affects the specific surface area of ZnO, the slow dripping process using a small needle often requires a very long reaction time. Therefore, we attempted to change the outlet needle and investigate the effect of the volumetric flow rate on the product's properties. The volumetric flow rates of the small needle, medium needle, large needle, and pipette were 0.0125, 0.020, 0.035, and 0.065 mL/s, respectively. As a result, the white precipitate was obtained 60°C, and the resulting product was examined using SEM, as shown in Figure 9. With regard to the microstructure, the ZnO produced with the small needle (a) and medium needle (b) can form nanorod shapes with very regular crystal morphology, with a width of about 50 nm and a length of over 400–500 nm. Some ZnO nanorods can even reach 1 μm in length. However, the crystal shape of ZnO prepared with the large needle (c) and pipette (d) is less regular, and the length of the nanorods is also significantly shorter. According to the surface area measuring instrument, the specific surface areas of ZnO prepared with the tiny needle, medium needle, large needle, and pipette are 15.6854, 14.1021, 13.0264, and 10.7432 m$^2$/g, respectively. It can be seen that the volumetric flow rate and specific surface area are inversely proportional, as shown in Figure 10. However, the reaction time required for preparing ZnO nanorods using the medium needle and pipette can be significantly reduced, although the volumetric flow

rate is larger. The original dripping time needed for the small needle was about 22.4 h, but this can be reduced to 4.3 h by replacing it with a pipette, which is of great help in preparing ZnO nanorods in a short time. Therefore, the direct precipitation method for preparing high-specific surface area ZnO nanorods by dripping is superior to a more significant volumetric flow rate. ZnO nanorods prepared under the control of four different volumetric flow rates were tested using XRD, with the results shown in Figure S1. The characteristic absorption peaks of ZnO crystalline structures were observed. This demonstrates that accurate control of the volumetric flow rate enables the synthesis of ZnO materials with nanostructures and shortens the time required to produce high-surface area ZnO nanorods. Moreover, different volumetric flow rates do not significantly impact the final yield.

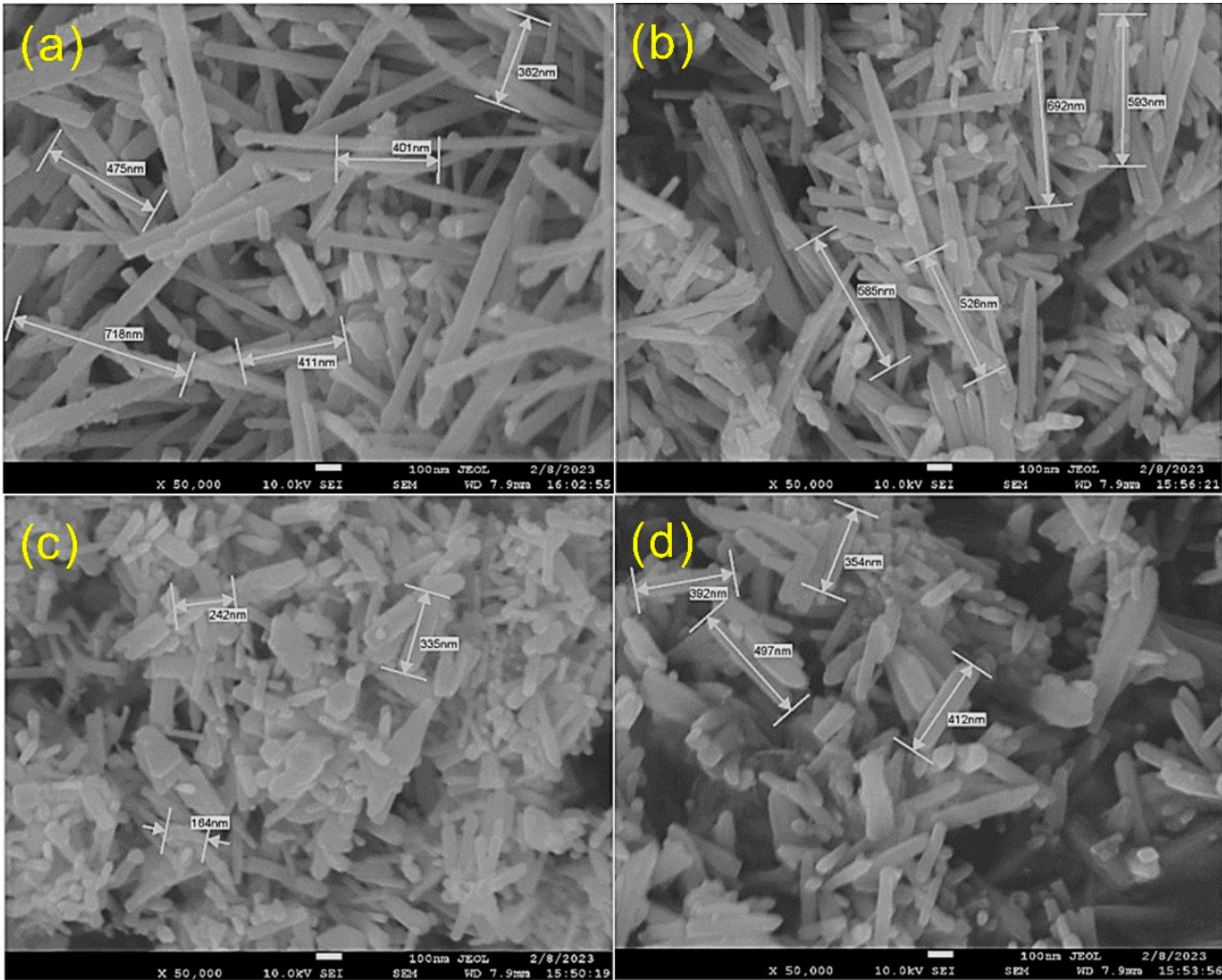

**Figure 9.** SEM images (50,000× magnification) of ZnO powder prepared by controlling the volumetric flow rate at (**a**) 0.0125 mL/s, (**b**) 0.020 mL/s, (**c**) 0.035 mL/s, and (**d**) 0.065 mL/s, while maintaining the reaction temperature at 60°C. The ZnO produced in (**a**,**b**) exhibited a nanorod morphology with regular crystalline shapes. The width was approximately 50 nm, and the length could reach over 400–500 nm, with some ZnO nanorods reaching up to 1 μm.

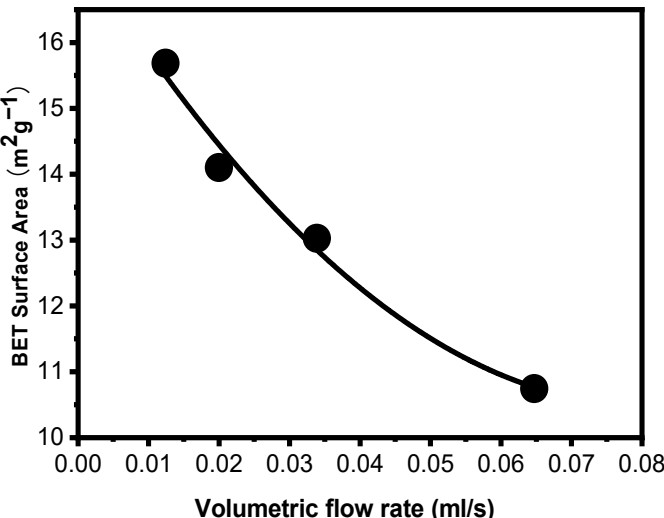

**Figure 10.** Comparison of the effect of different volumetric flow rates on the specific surface area of ZnO at a constant reaction temperature of 60 °C. The relationship between the volumetric flow rate and the specific surface area of ZnO exhibited an inverse proportionality trend.

*3.4. Comparison of Antibacterial Properties of Porous Diatomaceous Earth/ZnO Composites*

3.4.1. Comparison of Antibacterial Properties with Different Amounts of ZnO

DE composites, modified using inorganic materials and pore-forming agents, have porous properties and can maintain good strength in water. Therefore, they are suitable for antibacterial applications in a water environment. Using porous DE composite material as a carrier, 1%, 5%, and 10% ZnO (specific surface area of 15.6854 $m^2/g$) were added separately to the samples and compared with the control group (0% ZnO). The DE/ZnO composites were placed in a water environment containing only *E. coli* for growth, and the absorbance of the solution was measured using a spectrophotometer. As shown in Figure 11, the experimental results showed that although the DE composites have an extremely high specific surface area, they do not exhibit the characteristics of rapid adsorption and desorption of water vapor in a water environment. Instead, due to their particular pore structure, they become an environment for the proliferation of *E. coli*. Compared with other samples with added ZnO, the DE without ZnO added became turbid from the initially clear water solution due to the rapid proliferation of *E. coli*. Although the absorbance of the samples with added ZnO also increased gradually with the cultivation time, the absorbance of the DE with added ZnO was significantly lower than that of the DE composites without added ZnO. In particular, when the cultivation time reached 12 h, the absorbance ($OD_{600nm}$) of *E. coli* in DE without ZnO reached a peak value of the solution at 0.4771, while the absorbance ($OD_{600nm}$) of DE with 1%, 5%, and 10% ZnO added were 0.4535, 0.0428, and 0.0415, respectively. If the DE without ZnO added was used as the control group, the antibacterial rates of the three DE composites with added ZnO were 5.41%, 99.63%, and 99.93%, respectively. The experimental results showed that increasing the amount of ZnO added can improve the antibacterial rate. When the amount of ZnO added reaches 5% or more, the antibacterial rate against *E. coli* can be increased to over 99.63%.

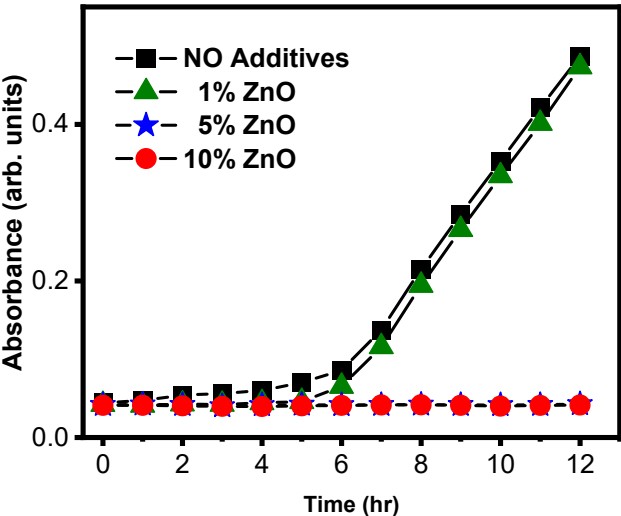

**Figure 11.** Comparison of the antibacterial effects of porous DE composites containing different proportions (0%, 1%, 5%, 10%) of nano-ZnO.

Although the ZnO particles prepared in this experiment had a high specific surface area, the antibacterial composites formed by mixing with DE showed that the ZnO powder distributed on the material's surface only accounted for a part of the entire material. Therefore, the antibacterial effect will be significantly reduced if the ZnO addition amount is insufficient. In this experiment, the antibacterial effect was not fully achieved when 1% ZnO was added. The *E. coli* could not be entirely in contact with ZnO to exert its antibacterial influence. As shown in Table 3, if the antibacterial experiment is continued for 24 h and then tested, the absorbance ($OD_{600nm}$) of the non-addition group and the 1%, 5%, and 10% ZnO addition groups will increase to 0.8323, 0.6674, 0.2860, and 0.1191, respectively. The DE composite material with 5% and 10% ZnO added initially, after undergoing long-term antibacterial testing in an aqueous environment, showed a slight increase in the absorbance of the culture medium, indicating that a small number of *E. coli* may be trapped between the glass wall of the beaker and the plastic net of the antibacterial composite material, and may not come into contact with ZnO, leading to the possibility of residual bacteria. Additionally, a dynamic equilibrium was formed under magnetic stirring. Nevertheless, from the experimental results, it can still be concluded that if the ZnO addition amount in the porous DE composites reaches above 5%, even under conditions of harsh water-phase environments and long-term testing, an antibacterial effect of more than 65% can still be achieved.

**Table 3.** Comparison of antibacterial rate in porous DE composites containing different proportions of nano-ZnO after 24 h.

| ZnO (%) | 0 | 1 | 5 | 10 |
|---|---|---|---|---|
| Absorption ($OD_{600nm}$) | 0.8323 | 0.6674 | 0.2860 | 0.1191 |
| Antibacterial rate (%) | — | 19.81 | 65.64 | 85.69 |

Growth temperature: 37 °C; detection wavelength: 600 nm.

### 3.4.2. Fluorescence Analysis of the Antimicrobial Experiment on DE Composites with Added ZnO

The experimental results and analysis, which were conducted to gain a deeper understanding of the actual growth of *E. coli* after treatment with DE/ZnO composite materials, are as follows. The *E. coli* culture, which had been cultivated for one day, was injected into the NB medium solution containing DE/ZnO composites for antibacterial experiments. The absorbance of the NB medium was tested every hour. After 12 h of antibacterial experimentation, 100 μL of the NB culture liquid was taken and stained with 5 μM SYTO

9 and 20 µM propidium iodide. After keeping in the dark for 15 min, the upper layer of dye was removed using a centrifuge. The bacteria were dissolved with 50 µL PBS solution, and 20 µL of the stained bacterial solution was taken and dropped onto a glass slide for observation under a fluorescence microscope.

Figures 12 and 13 show the results of the stained cultures without added ZnO composite materials and with 1% and 5% DE/ZnO composites under a microscope. From the photos, it is evident that there are significant differences between the *E. coli* cultures treated with zinc oxide-containing DE composite materials and those without. Figure 12 shows the experimental results without adding zinc oxide DE composites, with green fluorescent rod-shaped bacteria distributed significantly and less red fluorescence, indicating a higher number of live and fewer dead bacteria. However, when 1% zinc oxide was added to the DE composite materials, the fluorescence test results after 12 h showed a significant decrease in green fluorescent bacteria and a significant increase in red fluorescence, indicating that during the cultivation process, some *E. coli* came into contact with the nano-ZnO in the DE composite materials and died, as shown in Figure 13a.

| Phase Contrast | SYTO-Active *E. coli* | PI-Dead *E. coli* | Merge |
|---|---|---|---|

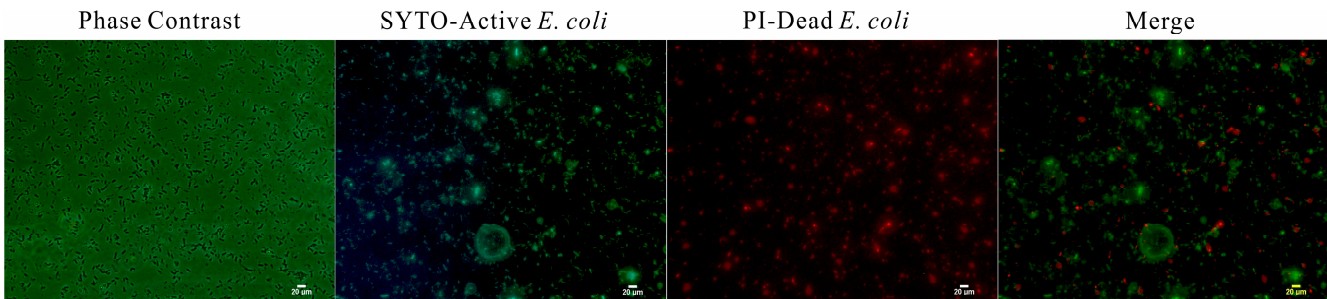

**Figure 12.** Using fluorescence microscopy for the fluorescence analysis of antimicrobial experiments on DE composites without added ZnO, we observed the growth trends of active and dead bacteria within 12 h.

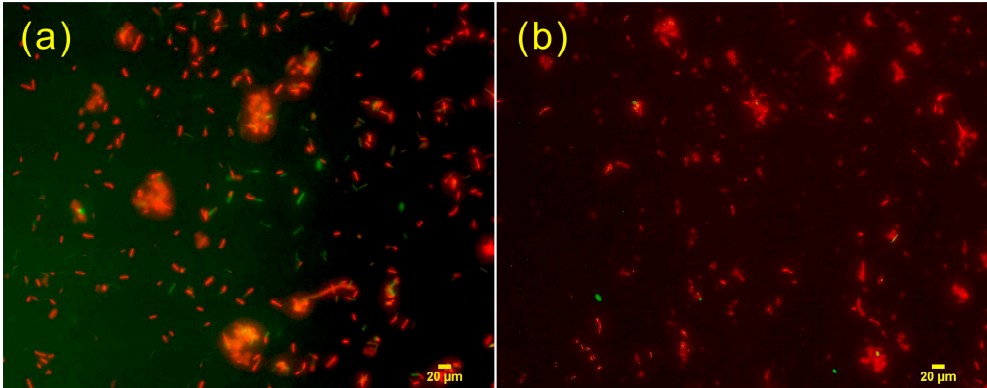

**Figure 13.** Fluorescence microscopy to observe DE composite materials with (**a**) 1% nano-zinc oxide and (**b**) 5% nano-zinc oxide added, studying the distribution of live (green) and dead (red) bacteria within 12 h (the nano-ZnO used had an average particle size of 72 nm).

The fluorescence observation results after 12 h In the presence of 5% zinc oxide in the DE/ZnO composites are shown in Figure 13b, with significant red fluorescence distribution. Nevertheless, the overall red fluorescence number is significantly lower than that of the group with 1% zinc oxide added. Furthermore, the green fluorescent rod-shaped bacteria in the image are rather scarce, indicating that during the bacterial cultivation process, due to the higher content of nano-ZnO in the DE composites, the chances of *E. coli* colliding or coming into contact with the nano-ZnO increased, making it difficult for the *E. coli* to grow in this environment. The image shows the experimental results of red fluorescent dead bacteria [54].

In the case of adding 10% ZnO to the DE composites, the results are similar to those with 5% ZnO added. Therefore, the results show that DE composites have a particular inhibitory effect on *E. coli*. Furthermore, the antibacterial effect increases with the nano-ZnO proportion added to the DE composites. These experimental results support further research on the antimicrobial applications of DE/ZnO composite materials.

### 3.4.3. Comparison of Antibacterial Properties of ZnO with Different Particle Sizes

The average particle sizes of ZnO powders synthesized at three temperatures of 25, 40, and 60 °C were determined through BET analysis. The particle sizes were 362, 103, and 72 nm, respectively. A 10% ZnO powder was mixed with porous DE composite material, and the resulting composite was tested for its antibacterial properties against *E. coli* in a liquid environment at 37°C. After 8 h of incubation in a water-soluble solution containing *E. coli* culture medium, it was observed that the solution soaked in the DE composite with added ZnO was more evident to the naked eye than the solution without ZnO. Samples were taken hourly from four groups with different particle sizes of ZnO added to the DE composite. The absorbance values of the culture medium were measured using a spectrophotometer, as shown in Figure 14. DE composite with 72 and 103 nm ZnO powders added showed no significant changes in absorbance over time, while DE composite with 362 nm ZnO powder added and the control group showed a steep increase in absorbance after 5 h, which gradually slowed down by the seventh hour, approaching a saturation state. As shown in Table 4, the absorbance values of the culture medium without ZnO and with 72, 103, and 362 nm ZnO, measured after 8 h, were 0.4091, 0.0209, 0.0224, and 0.3701, respectively. Using DE composite as the control group, the antibacterial rates against *E. coli* were 94.89%, 94.52%, and 9.53% for the three groups with different ZnO particle sizes below 100 nm, indicating that ZnO particles with a size below 100 nm exhibit excellent antibacterial effects against *E. coli*. ZnO synthesized at 60°C showed a nanorod morphology. Its small size and rod-like shape allowed it to pierce the bacterial cell membrane and cell wall upon contact with bacteria, leading to bacterial death [34]. In addition, ZnO nanorods can absorb ultraviolet light, producing electron–hole pairs, which can generate redox reactions. When *E. coli* comes into contact with ZnO nanorods, electrons and holes are developed on their surface, having some free radicals, which can oxidize organic matter on the cell membrane and cell wall, thereby inhibiting the growth of *E. coli* [33]. ZnO synthesized at 40°C showed a nanosphere morphology and could release zinc ions in water, which could act as antibacterial agents to inhibit the growth of *E. coli*. Zinc ions can interfere with enzyme activity on the cell membrane and cell wall, destroying the cell structure and metabolic processes and inhibiting bacterial growth [35]. The size of *E. coli* is approximately $0.5 \times (1{\sim}3)$ μm, which is more than ten times larger than the three sizes of ZnO powders prepared in this experiment. This study synthesized nano-sized ZnO particles, where the specific surface area of these nanoparticles is inversely proportional to their particle diameter. The smaller-sized ZnO particles not only significantly increase the likelihood of collisions with *E. coli* but also contribute to the photocatalytic activity and redox reactions of $Zn^{2+}$. If the particles are in the form of nanorods, a mechanical puncturing effect is generated, further enhancing the antibacterial performance of the DE composite material.

**Table 4.** Antibacterial rate of ZnO with different particle sizes added to porous DE composites against *E. coli*.

| ZnO $d_{\mathrm{BET}}$ (nm) | 0 | 72 | 103 | 362 |
|---|---|---|---|---|
| Absorption (OD$_{600nm}$) | 0.4091 | 0.0209 | 0.0224 | 0.3701 |
| Antibacterial rate (%) | - | 94.89 | 94.52 | 9.53 |

Growth temperature: 37 °C; detection wavelength: 600 nm.

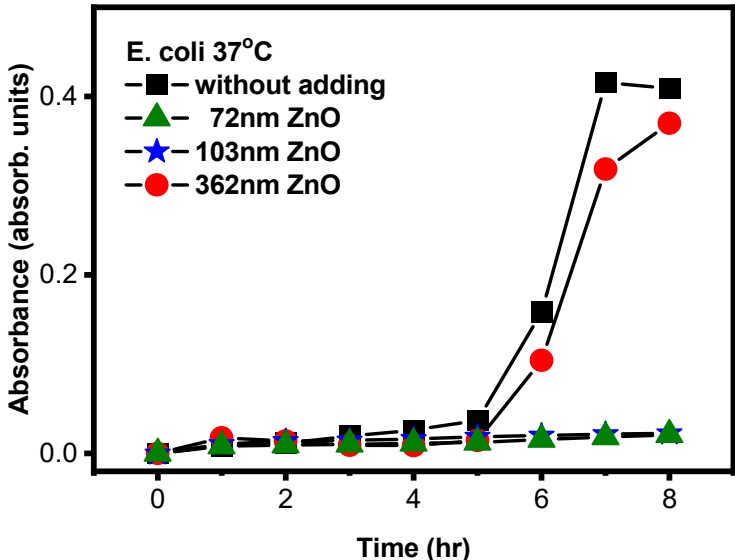

**Figure 14.** Comparison of the antibacterial efficiency of porous DE composite materials containing 10% ZnO with different particle sizes against *E. coli*. The smaller-sized nano-ZnO exhibited a better inhibitory effect on the growth of *E. coli*.

### 3.5. Practical Application of Porous Diatomaceous Earth/Nano-ZnO Composites

Based on the above research results, it was found that DE, through the addition of inorganic materials, pore-forming agents, and nano-ZnO antibacterial materials, not only significantly improved the strength of DE in water but also the composite material formed after molding due to its porous nature, which can increase the antibacterial rate of internal nano-ZnO. Using a self-made DE mold formula combined with a commercially available silicone rubber mold (silicone rubber and curing agent), a DE antibacterial composite material suitable for different landscapes in fish tanks can be successfully produced. The molded product is shown in Figure S2, with the following descriptions: (a) Roman arena; (b) lattice fence; (c) owl flowerpot; (d) fox decoration; (e) doll; (f) lantern landscape decoration. The finished product illustrations show that different appearances can be easily shaped using simple molding techniques. In addition, inorganic pigments can be added or coated on to improve visual appeal according to actual needs. Finally, the molded products were placed in a fish tank for landscaping and testing, and the actual results after one month of water testing are shown in Figure 15. The landscaping decorations in the fish tank showed no signs of disintegration. If the DE composite material is made into a lattice shape (as shown in Figure S1b), it can also be used as a pre-filtering material in a fish tank. When circulating water passes through the porous DE antibacterial material formed by the multi-layer combination lattice shape, it can improve water quality through its antibacterial effect and reduce the chance of fish disease. Compared with traditional fish tank landscaping decorations, using DE as a base to produce antibacterial composite material suitable for water-based landscaping has higher plasticity and the antibacterial effect of inhibiting bacterial growth in fish tanks and reducing water replacement problems. Therefore, the application and development of this material are highly promising.

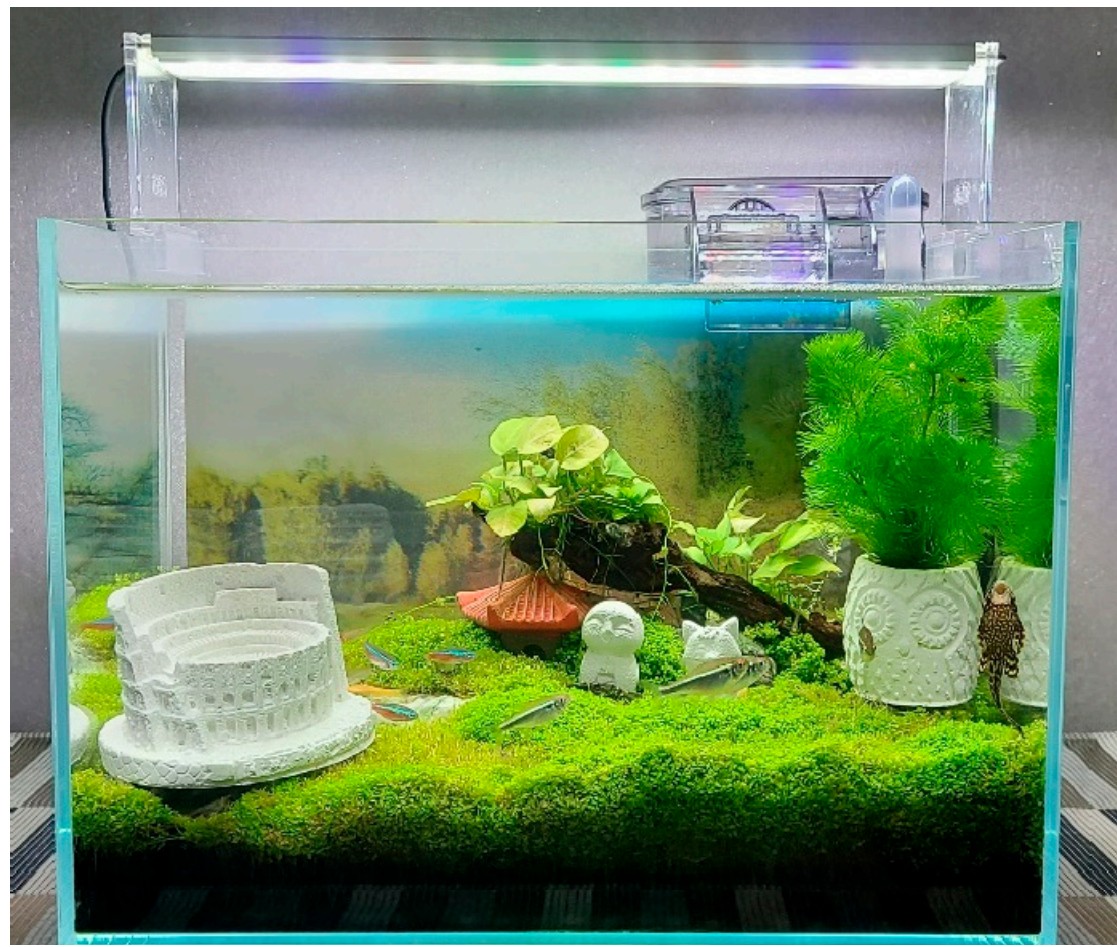

**Figure 15.** Application of self-made porous DE/nano-ZnO composites for antibacterial purposes in an aquarium.

## 4. Conclusions

The present study used DE, inorganic TST gypsum, and white cement powder to prepare DE composite material. The strength of DE after molding was successfully improved, and the drying time was shortened. DE can undergo hydration when mixed with cement powder, significantly increasing the strength of the DE after molding. The longer the time, the greater the strength. When the weight ratio of DE to white cement powder is 4:6, the molded DE has both mechanical strength and water absorption. Adding 20–30% TST gypsum can significantly improve the DE's drying and molding rate. In the experiment of making porous DE composite material, it was found that after adding $2Na_2CO_3 \cdot 3H_2O_2$, the DE composite material had a porous cheese-like structure after washing with water, and adding NaCl made the original composite material produce smaller pore structures. Both pore-forming agents can dissolve during washing to form pore structures inside the DE composite material.

Using an improved direct precipitation method, the specific surface area of the ZnO powder prepared using titration was greatly affected by the reaction temperature conditions, followed by the volume flow rate. Within 25–60 °C, a higher reaction temperature and lower volume flow rate of KOH and $ZnCl_2$ can yield ZnO powder with a larger specific surface area. When the reaction temperature is increased from 25 °C to 60 °C, the specific surface area of the powder is increased by about five times, and the average particle size of ZnO is reduced from 362 nm to 72 nm. Furthermore, it was found that when the reaction temperature was maintained at 60 °C, the crystalline form of ZnO changed from a spherical to nanorod shape. Using a micro-needle to control the flow rate at 0.02 mL/s can help the growth of nanorods, and the nanorods can reach more than one micrometer. As ZnO

nanorods have excellent optical properties and are highly sensitive to gases, temperature, pressure, humidity, biological molecules, etc., this discovery will be helpful for future applications in solar cells and various sensors.

Porous DE composite material does not have antibacterial properties against *E. coli*. However, after adding 1, 5, and 10% of 68 nm ZnO nanorods, the antibacterial rate against *E. coli* can reach 5.41%, 99.63%, and 99.93% for 12 h, respectively. This low-cost and simple operation is an experimental method to prepare porous DE/nano-ZnO composite material. It can be used to prepare materials with antibacterial, dehumidification, and filtration applications without using high-standard experimental equipment. This porous material can be applied in various fields, such as water purification, air filtration, and food packaging.

**Supplementary Materials:** The following supporting information can be downloaded at: https://www.mdpi.com/article/10.3390/jcs7050204/s1, Figure S1: XRD crystal diffraction analysis of nano-ZnO at different the volumetric flow rate: (a) 0.0125 mL/s; (b) 0.020 mL/s; (c) 0.035 mL/s; (d) 0.065 mL/s.; Figure S2: Replicated products of self-made porous DE/nano ZnO composite materials.

**Author Contributions:** Conceptualization, J.-J.H. and C.-C.C.; methodology, J.-J.H. and C.-C.C.; formal analysis, J.-J.H.; investigation, J.-J.H. and C.-C.C.; resources, C.-C.C.; data curation, J.-J.H.; writing—original draft preparation, J.-J.H. and C.-C.C.; writing—review and editing, C.-C.C. and J.-J.H.; visualization, J.-J.H. All authors have read and agreed to the published version of the manuscript.

**Funding:** This research received no external funding.

**Data Availability Statement:** The data that support the findings of this study are available from the corresponding author upon request.

**Acknowledgments:** The authors sincerely appreciate the guidance and support provided by Lin Shu-Rung of the Department of Bioscience Technology at Chung Yuan Christian University in bacterial culture equipment and techniques. Likewise, we would like to express our heartfelt gratitude to Professor Yeh Jui-Ming and Li Min-Xue of the Department of Chemistry at Chung Yuan Christian University for their invaluable assistance and support in instrument equipment and material testing. Furthermore, we sincerely thank Ma Te-Wei of the Department of Chemical Engineering at the Army Academy for his generous help providing bacterial strains and assisting with bacterial culture techniques.

**Conflicts of Interest:** The authors declare no conflict of interest.

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
