# Peer review of "Porous Diatomaceous Earth/Nano-Zinc Oxide Composites: Preparation and Antimicrobial Applications"

_jcs, doi:10.3390/jcs7050204_

Round 1
Reviewer 1 Report
This manuscript presents the preparation of a porous diatomaceous earth (DE) composite material and a high surface area ZnO nanoparticles (NPs), and their mixing with 5% ZnO nanorods to enhance the antibacterial properties of ZnO NPs. The resulting composite showed up to a 100% rate of antibacterial activity against E. coli in an aqueous environment.
The authors studied various chemico-physical phenomena of the composites and characterized their properties, including adding inorganic materials and white cement to improve DE strength and drying rate, dissolving sodium percarbonate or sodium chloride to create different pore structures of DE, and controlling temperature, volume flow, and titration rate to increase the specific surface area of ZnO NPs.
Although the preparation methods of DE and ZnO NPs are not novel, and the antibiotic/antibacterial properties of ZnO NPs are well known, the authors have contributed a relatively simple and low-cost way of fabricating the composite material, with a very detailed description of the processes. This can serve as a reference and be helpful to researchers working in this field.
The manuscript is well-organized and well-written, and I recommend its publication in its current form.
However, it would be helpful if the authors could explain the mechanism behind why sodium percarbonate or sodium chloride creates a different pore structure of the composite, and why the antibacterial reactivity of ZnO NPs can be enhanced with an increase in its specific surface area.
Author Response
Dear Reviewer:
We are immensely grateful to the reviewer for their invaluable guidance and endorsement of our manuscript. Their insightful comments have greatly encouraged us, and we have addressed the two questions raised by the reviewer with the following responses, which have also been incorporated into our revised paper.
Firstly, we have supplemented the explanation at the end of Section 3.2, "Comparison of Porosity of DE Composites," on why sodium percarbonate or sodium chloride would result in different pore structures within the composite materials.
Additional information as follows:
Both sodium chloride and sodium percarbonate can be used as pore-forming agents for diatomaceous earth composites. Removing the pore-forming agent dispersed in the composite material using a dissolution method is a simple and environmentally friendly method. If sodium chloride is used as the pore-forming agent, the pore structure in the composite will be related to the size of the mixed sodium chloride crystals. Strictly controlling the solubility of sodium chloride in water can retain the residual sodium chloride crystals in the composite material. Immersing the diatomaceous earth composites in hot water can dissolve the sodium chloride crystals and naturally form a large number of uniform porous structures without affecting the original strength of the composite material. On the other hand, when sodium percarbonate is used, it releases a large amount of oxygen when it comes into contact with water. Since the oxygen is encapsulated in the diatomaceous earth composites, the originally dense structure will form a loose structure due to the encapsulation of a large amount of oxygen. The formation of pores comes from the oxygen encapsulated in the composite material and the remaining sodium percarbonate particles, and the pore size is less uniform.
Secondly, at the end of Section 3.3.2, "Comparison of antibacterial properties of ZnO with different particle sizes," we have added further clarification on why the antibacterial reactivity of zinc oxide NPs can be enhanced with an increase in their specific surface area.
Additional information as follows:
This study synthesized nano-sized ZnO particles, where the specific surface area of these nanoparticles is inversely proportional to their particle diameter. The smaller-sized ZnO particles not only significantly increase the likelihood of collisions with E. coli but also contribute to the photocatalytic activity and redox reactions of Zn2+. If the particles are in the form of nanorods, a mechanical puncturing effect is generated, further enhancing the antibacterial performance of the DE composite material.
Reviewer 2 Report
1、The author should confirm the sterilization conditions “100°C for 15 min”, while the common conditions are 121°C for 15 min;
2、OD values cannot quantitatively evaluate the antibacterial activity, while it can only reflect the number of bacteria, including the live and dead ones;
3、Scale bar should be incorporated in the images of Fig. 5;
4、In Conclusions,delete “innovative”, the way presented is a common way to prepare composites;
5、How to ensure the uniform distribution of inorganic TST gypsum, 2Na2CO3•3H2O2, ZnO inside the DE based composites, the authors should clarify them;
6、As the authors stated in the title, Preparation and Antimicrobial Applications, however, the authors only conducted a few antibacterial experiments, and the presented data are superficial, thus the relevant study should be enriched.
Author Response
Dear Reviewer:
We are grateful to the reviewer for their valuable suggestions and comments on our paper, which have been very encouraging to us. We would like to respond to the questions raised by the reviewer and have incorporated the results into our revised paper accordingly.
- The author should confirm the sterilization conditions“100°C for 15 min”, while the common conditions are 121°C for 15 min;
Reply: (In Experimental Section 2.5, we have added the following explanation and corrections.)
- The sterilization conditions mentioned in the article were initially set at 100°C for 15 minutes, but have since been corrected to 100°C for 30 minutes.
- Generally, the higher the temperature, the shorter the required sterilization time. Sterilization can be achieved by processing materials in saturated high-pressure steam at 121°C for 15 minutes. However, considering the purpose of this study, the sterilization of Nutrient Broth (NB) medium and composite materials was performed using a magnetic stirrer to directly heat and boil them continuously for 30 minutes. The NB medium solution was subjected to these sterilization conditions and left at room temperature for 3 days. The changes in absorbance of the culture medium during this period were examined using a spectrophotometer, and the experimental results confirmed that this heating method can also achieve good sterilization efficacy.
2、OD values cannot quantitatively evaluate the antibacterial activity, while it can only reflect the number of bacteria, including the live and dead ones;
Reply: By measuring the turbidity of bacterial solutions using optical density (OD600nm), we can roughly infer the concentration of bacteria and understand the trend of bacterial growth or the effect of antibacterial agents. However, this method indeed cannot differentiate between the number of live and dead bacteria in the solution. We will address this issue in our response to the sixth question.
3、Scale bar should be incorporated in the images of Fig. 5;
Reply: Thanks to the reviewer's reminder, we have included a scale bar in each photo in Figure 5 to facilitate the reader's understanding of the size and shape of the holes.
4、In Conclusions,delete “innovative”, the way presented is a common way to prepare composites;
Reply: Thanks to the reviewer's suggestion, we have removed the word “innovation” in the Conclusion for innovative experiments.
5、How to ensure the uniform distribution of inorganic TST gypsum, 2Na2CO3•3H2O2, ZnO inside the DE based composites, the authors should clarify them;
Reply: (We add a description of how to uniform the composites to the end of Experimental section 2.5.)
According to the guidelines of the United States Pharmacopoeia (USP) and the European Pharmacopoeia (EP), the blending process should be carried out using suitable equipment and adequate blending should be carried out within a minimum of 15 min to a maximum of 60 min. In order to achieve uniform distribution of inorganic TST gypsum, 2Na2CO3-3H2O2 and ZnO in DE-based composites, the four powders were packed in well-sealable plastic can and operated continuously for 30 minutes using a V-shaped blending equipment to ensure adequate blending of the powders for subsequent experiments.
6、As the authors stated in the title, Preparation and Antimicrobial Applications, however, the authors only conducted a few antibacterial experiments, and the presented data are superficial, thus the relevant study should be enriched.
Reply:
First and foremost, we would like to express our gratitude to the reviewer for their invaluable suggestions. This paper primarily presents a simple and cost-effective method for synthesizing nano-sized ZnO. Given its antibacterial properties, we utilized a blending technique to combine it with diatomaceous earth composite materials, successfully creating highly structurally stable ZnO/DE composite materials, which is the crux of our research.
Throughout the antibacterial (E. coli) experiments, we carefully observed the positive antimicrobial effects of ZnO/DE composites in aqueous environments. However, we also acknowledge that our research on antibacterial aspects is not yet comprehensive. Bacterial experimentation requires stringent cultivation methods, procedures, and reproducibility, all of which demand considerable time. Nevertheless, we find the reviewer's suggestions constructive and have already embarked on additional antibacterial experiments.
In our next paper, we plan to use the ZnO/DE composites to conduct further antibacterial efficacy research, comparisons, and analyses on E. coli, S. aureus, P. aeruginosa, and various fungi. Our primary research methods will encompass two approaches: firstly, using the plate count method to calculate bacterial colonies, obtaining the quantity of colony-forming units (CFUs) per milliliter in the original solution; secondly, employing selective staining of live (SYTO 9) or dead (propidium iodide, PI) bacteria with fluorescent dyes and utilizing fluorescence microscopy to observe and differentiate between live and dead bacteria. Through these methods, we will be able to accurately quantify bacterial growth in solution and the difference in numbers between live and dead bacteria. We believe that, in the future, we will be able to provide more comprehensive results and analytical reports.
Round 2
Reviewer 2 Report
The additional antibacterial results and the corresponding analysis should be incorporated into the revised version.
Author Response
Reviewer’s suggestion: The additional antibacterial results and the corresponding analysis should be incorporated into the revised version.
Dear Reviewer:
We are extremely grateful to the reviewer for the valuable suggestions provided on our paper. Although considerable time was spent conducting bacterial culture experiments and tests, the results were indeed exciting. We believe these will enhance the credibility of our paper, particularly in the aspect of antimicrobial properties of materials. We have incorporated the relevant results and analyses into our revised paper, with the supplemental results as follows:
Reply:
- We hereby supplement with the 4.2 bacterial fluorescence experimental section and provide Figures 12 and 13, which display the growth trends of the bacteria.
- (Acknowledgments) We would also like to extend our special thanks to other research teams for their assistance. (Page# 28)
- We include the model and specifications of the fluorescence microscope used in our study. (Supplement in Section 2.2, Instrumentation)
3.4.2 Fluorescence analysis of the antimicrobial experiment on DE composites with added ZnO
The experimental results and analysis, which were conducted to gain a deeper understanding of the actual growth of E. coli after treatment with DE/ZnO composite materials, are as follows. The E. coli culture, which had been cultivated for one day, was injected into the NB medium solution containing DE/ZnO composites for antibacterial experiments. The absorbance of the NB medium was tested every hour. After 12 hours of antibacterial experimentation, 100μL of the NB culture liquid was taken and stained with 5μM SYTO 9 and 20μM propidium iodide. After avoiding light for 15 minutes, the upper layer of dye was removed using a centrifuge. The bacteria were dissolved with 50μL PBS solution, and 20μL of the stained bacterial solution was taken and dropped onto a glass slide for observation under a fluorescence microscope.
Figures 12 and 13 show the results of the stained cultures without added ZnO composite materials and with 1% and 5% DE/ZnO composites under a microscope. From the photos, it is evident that there are significant differences between the E. coli cultures treated with zinc oxide-containing DE composite materials and those without. Figure 12 shows the experimental results without adding zinc oxide DE composites, with green fluorescent rod-shaped bacteria distributed significantly and fewer red fluorescence, indicating a higher number of live and fewer dead bacteria. However, when 1% zinc oxide was added to the DE composite materials, the fluorescence test results after 12 hours showed a significant decrease in green fluorescent bacteria and a significant increase in red fluorescence, indicating that during the cultivation process, some E. coli came into contact with the nano-ZnO in the DE composite materials and died, as shown in Figure 13(a).
In the presence of 5% zinc oxide in the DE/ZnO composites, the fluorescence observation results after 12 hours are shown in Figure 13(b), with significant red fluorescence distribution. Still, the overall red fluorescence number is significantly lower than that of the group with 1% zinc oxide added. Furthermore, the green fluorescent rod-shaped bacteria in the image are pretty scarce, indicating that during the bacterial cultivation process, due to the higher content of nano-ZnO in the DE composites, the chances of E. coli colliding or coming into contact with the nano-ZnO increased, making it difficult for the E. coli to grow in this environment. The image shows the experimental results of red fluorescent dead bacteria.
In the case of adding 10% ZnO to the DE/ZnO composites, the results are similar to those with 5% ZnO added. Therefore, the results show that DE/ZnO composites have a particular inhibitory effect on E. coli. Furthermore, the antibacterial effect increases with the nano-ZnO proportion added to the DE/ZnO composites. These experimental results support further research on the antimicrobial applications of DE/ZnO composite materials.
Figure 12. Using fluorescence microscopy for the fluorescence analysis of antimicrobial experiments on DE composites without added ZnO, we observed the growth trends of active and dead bacteria within 12 hours.
Figure 13. Fluorescence microscopy to observe DE composite materials with added (a) 1% nano-zinc oxide and (b) 5% nano-zinc oxide, studying the distribution of live (green) and dead (red) bacteria within 12 hours. (The nano-ZnO used had an average particle size of 72 nm)
Acknowledgments
The authors sincerely appreciate the guidance and support provided by Professor Lin Shu-Rung of the Department of Bioscience Technology at Chung Yuan Christian University in bacterial culture equipment and techniques. Likewise, we would like to express our heartfelt gratitude to Professor Yeh Jui-Ming and Mr. Li Min-Xue of the Department of Chemistry at Chung Yuan Christian University for their invaluable assistance and support in instrument equipment and material testing. Furthermore, we sincerely thank Professor Ma Te-Wei of the Department of Chemical Engineering at the Army Academy for his generous help providing bacterial strains and assisting with bacterial culture techniques.
The model and specifications of the fluorescence microscope
All reagents were of reagent grade unless otherwise stated E. coli that fluoresces when stimulated by UV light was observed using a fluorescent microscope (Nikon TE2000-U, Japan). and the filter used for autofluorescence detection of E. coli is UV (330~380 nm, Nikon).

Round 3
Reviewer 2 Report
None